# Artifact Readiness Gates with Saturation Stop Rules and Host-Parity Admissibility for FM Release Evaluation

Yanick Kanyiki
InvarLock
Ottawa, Canada
yanick.kanyiki@invarlock.ai

## ABSTRACT

Release evaluation for FM-powered software often grows by habit rather than policy: teams repeat runs until budget or time is exhausted, without clear evidence that more passes change release decisions. We study a release-evaluation protocol that separates three concerns: artifact readiness, decision-stability stopping, and cross-hardware promotion gating. The study uses 340 runs spanning seven edit families (five core plus two probes), four model families, ten seeds, and dual-host H100/H200 execution. In this matrix and under this policy setting, additional seed repetition did not change promote/block outcomes, edit-family breadth remained decision-informative, and small H100/H200 score differences could still alter promotion outcomes near strict boundaries. These findings motivate workload-conditional resource allocation for release engineering: in this evidence setting, additional budget is more decision-informative when spent on edit diversity and host-parity checks than on deeper seed repetition. The contribution is an operational decision framework, with explicit sensitivity reporting, that turns release evaluation from a fixed checklist into a defensible governance process. In this matrix, seed-stop reduced measured GPU-hours by about 90% versus fixed 10-pass seed evaluation. Numeric thresholds are workload-derived; the transferable contribution is the gate-setting process.

## CCS CONCEPTS

• **Software and its engineering** → **Software creation and management**; • **Computing methodologies** → *Machine learning*.

## KEYWORDS

AI-powered software, release engineering, trustworthiness, evaluation gates, host parity, saturation stop rules, InvarLock

**ACM Reference Format:**
Yanick Kanyiki. 2026. Artifact Readiness Gates with Saturation Stop Rules and Host-Parity Admissibility for FM Release Evaluation. In *Proceedings of the 3rd ACM International Conference on AI-powered Software (AIware '26), July 6–7, 2026, Montreal, QC, Canada.* ACM, New York, NY, USA, 9 pages. https://doi.org/TBD

## 1 INTRODUCTION

Release pipelines for foundation model (FM)-powered software increasingly look like experiment grids: many edits, multiple stages, and repeated seeds. In practice, teams still lack defensible answers to two questions: when is the artifact set clean enough to trust, and when do additional evaluations stop changing release decisions. A common failure mode is over-evaluation without added decision value, for example running a full 10-seed-by-multi-edit matrix while only a small subset of those runs actually changes promote/block outcomes.

This gap is operationally expensive. In release settings with fixed deployment windows, unbounded evaluation depth either delays release decisions or forces late-stage truncation without rationale. Both outcomes reduce trust in the release process: teams either ship late despite stable evidence or ship on incomplete evidence without an auditable stopping rule. Applied release engineering therefore needs policy-level criteria for "enough evidence," not only larger evaluation matrices.

A representative failure pattern from this study context is a team executing a full 10-seed by 5-edit matrix on dual hosts, then finding after completion that extra seed passes did not alter final decisions while cross-host checks still blocked boundary promotions. The practical question is not whether more runs can be executed, but which additional runs are most likely to change release decisions under explicit policy semantics.

This paper addresses that policy gap with an operational release-evaluation framework centered on three gates: artifact readiness, evidence-depth stopping, and host-parity admissibility. We treat release evaluation as a governance problem rather than a benchmark maximization problem. The study asks three applied questions: does seed repetition materially change decisions, does edit-family diversity add decision signal, and do cross-host differences affect promotion outcomes. The contribution is therefore about release-policy outputs and auditability, not about improving downstream task quality.

*Contributions.*

- **A staged release-governance protocol.** We formulate artifact readiness, evidence-depth stopping, and host-parity admissibility as separate gates with auditable policy outputs.
- **Sensitivity-aware stopping analysis.** We report where additional passes stopped changing promote/block outcomes in this matrix, together with explicit order and missingness sensitivities.
- **Cross-host admissibility evidence.** We quantify how parity thresholds shift admissibility and promotion outcomes across H100 and H200 under symmetric no-parity baselines.

- **Deterministic multi-model evidence and scoped guidance.** We report results from 340 runs over multiple model families, edit families, seeds, and H100/H200 hosts, with deterministic artifact regeneration and workload-conditional operational guidance.

## 2 RELATED WORK

Production ML guidance emphasizes deployment hygiene, observability, and lifecycle debt management [3–5, 9, 19]. Complementary release-engineering studies map CI/CD practices, rollout tradeoffs, and organizational constraints in modern delivery settings [1, 11, 17, 18, 20]. Staged release practice motivates canary-first controls [6], and runtime verification/monitoring literatures contribute operational checking concepts [12]. Our contribution differs by making release-decision governance, rather than deployment infrastructure, the primary object of study.

Stopping decisions are also connected to sequential analysis. Classic sequential-test framing asks when incremental evidence no longer changes a decision boundary [23]. Our stop-depth metric is operational rather than inferential: it tracks promote/block convergence under fixed policy semantics and then measures order sensitivity via pass permutations. The focus is decision stabilization under practical release constraints, not hypothesis testing under distributional assumptions.

Cross-device reproducibility is a practical concern in modern ML release stacks. Determinism guidance from hardware/software vendors highlights that numerics can vary across kernels, driver stacks, and runtime configurations [15], and reproducibility guidance in ML emphasizes explicit reporting and replay contracts for credible claims [16]. We translate that concern into a release-governance control: host-parity admissibility with explicit thresholds and blocked-promote accounting. This reframes reproducibility from a diagnostics question into an explicit promote/block gate.

LLM benchmarking and regression tooling (including HELM-style breadth and CI-oriented prompt checks) provide broad capability and regression signals [2, 13, 22]. Structured reporting practices such as model cards motivate explicit disclosure contracts that can be adapted to release-decision reporting [14]. Our focus is narrower and release-policy specific: admissibility, stopping, and integrity as first-class decision semantics.

## 3 APPROACH

### 3.1 Protocol Overview

The release protocol applies three stages in sequence: artifact readiness, evidence-depth stopping, and parity-gated promotion. Before analysis, an artifact readiness check validates report parsing, model identity, baseline identity, runtime payload, and metric finiteness. Rows with error-level issues block analysis; warning-level rows remain visible evidence. This keeps data-quality failures separate from release-policy outcomes.

The readiness gate tracks issue classes because remediation differs by failure type. Identity mismatches require provenance correction and rerun; parse/runtime payload failures require pipeline repair; non-finite metrics require model-edit diagnosis and guarded exclusion rules. Even when error-level issues are absent in a final

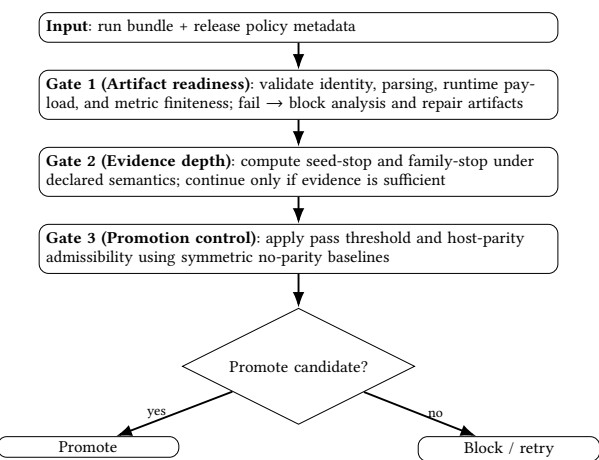

**Figure 1: Three-stage release-gating workflow used in this study.**

bundle, retaining this taxonomy clarifies why the gate is a prerequisite rather than a downstream quality score. This separation is consistent with broader ML production-readiness and reporting guidance that emphasizes explicit validation and traceable review artifacts [5, 14, 16].

InvarLock provides the typed evaluation artifacts used by this protocol: per-run reports with canary/deployment quality scores, integrity diagnostics, and policy metadata keyed by deterministic run identifiers [7, 8]. Our analysis consumes those reports as inputs and derives release-policy outputs; it does not modify underlying model evaluation metrics. The protocol is portable at the interface level in principle: a runner that emits the same minimal fields (canary/deployment scores, integrity status, host tags, and deterministic run keys) can implement the same gates. This paper, however, evaluates only one toolchain and does not empirically validate cross-runner replication. Figure 1 summarizes this three-stage flow.

> **Terminology box. Artifact readiness** checks whether the evaluation bundle is valid enough to analyze; **seed-stop** asks when more seed repetitions stop changing the release decision; **family-stop** asks when added edit-family coverage stops changing that decision; and **host parity** asks whether the same candidate remains admissible across the studied hardware pair before promotion.

### 3.2 Saturation Stop Rules

The central stopping question is: how many passes are needed before promote/block decisions stabilize? We measure this with stop depth. For pass $t$, define row-level pass indicator $p_t = 1[c_t \geq \tau \wedge u_t \geq \tau]$, where $c_t$ and $u_t$ are canary and deployment quality scores and $\tau$ is the strict pass threshold. The cumulative release decision after pass $t$ is

$$d_t = \begin{cases} \texttt{promote}, & \text{if } \bigwedge_{i=1}^{t} p_i = 1 \\ \texttt{block}, & \text{otherwise.} \end{cases}$$

Stop depth is then

$$n^* = \min\{t \mid d_j = d_T, \ \forall j \geq t\},$$

where $T$ is full evidence depth.

We evaluate two axes. **Seed-stop** adds seeds for fixed edit, model, and host. **Family-stop** adds edit families for fixed model and host. Family-stop uses two explicit semantics: observed coverage (only measured families) and conservative coverage (global family list with missing-as-fail). This separation avoids conflating unrun probe cells with empirical failure. For operational use, family order should be declared a priori; a practical heuristic is cost-first triage from low-cost, high-yield families toward higher-cost families until stabilization.

Because stop depth depends on ordering, we also compute permutation summaries (P10/P50/P90) and compare them with the ordered trajectory per group.

### 3.3 Host-Parity Admissibility

When the same edit is evaluated on different hardware, small score differences can still flip decisions near strict thresholds. We model this with host-parity admissibility. For matched H100/H200 rows with the same edit, model, and seed:

$$\Delta_{host} = \max\left(|c_{100} - c_{200}|, |d_{100} - d_{200}|\right).$$

A pair is admissible if $\Delta_{host} \leq \epsilon$. Promotion with parity requires both host decisions to pass and the pair to be admissible. To avoid baseline bias, we report blocked-promote counts against three no-parity baselines: H100-only, H200-only, and Any-host-pass.

## 4 EVALUATION DESIGN

### 4.1 Toolchain and Hardware

InvarLock is pinned to v0.3.10 and documented contracts [7, 8]. Execution uses one dual-H100 host and one dual-H200 host. All analysis artifacts are generated by scripts, not hand-edited tables.

### 4.2 Confirmatory Core Matrix

The core matrix includes 300 rows (150 H100/H200 pairs):

- models: Qwen2.5-7B, Qwen2.5-14B, and Mistral-7B.
- edits: 8-bit quantization (RTN compression), 4-bit quantization (more aggressive RTN compression), a LoRA-like rank-8 edit (parameter-efficient adapter perturbation), runtime prompt wrapping (input-template intervention), and inference-config perturbation (decoder setting change).
- seeds: 43–52.
- stages: canary and deployment on both hosts.

### 4.3 Targeted Stress Probes

The evaluation design uses two tiers: a confirmatory core matrix for primary claims and targeted probes for boundary stress:

- **External-family probe**: TinyLlama-1.1B, dual host, 10 seeds (20 calibration rows; 40 stage runs).
- **Controlled stochasticity probe**: Mistral-7B with a stochastic quantized preset, dual host, 10 seeds (20 calibration rows; 40 stage runs), fixed policy/configuration, and seed-only variation across runs.

**Table 1: Evaluation matrix coverage summary.**

| Bundle | Rows | Host pairs | Composition (edits/models/seeds) |
|---|---|---|---|
| Core matrix | 300 | 150 | 5/3/10 |
| Augmented (core + probes) | 340 | 170 | 7/4/10 |

These probes are targeted stress tests; they do not claim full Cartesian coverage across all models and edits. Coverage totals and compact composition counts are shown in Table 1.

### 4.4 Statistical Reporting and Reproducibility

The run matrix is deterministic by construction (fixed manifests and seeds), so bootstrap intervals can collapse. We therefore report spread statistics (min/P50/P90/max) alongside bootstrap CI fields to avoid overstating certainty from deterministic replay, following broader reproducibility-reporting expectations in ML research [16]. This paper uses 200 permutation samples per group and 2,000 bootstrap resamples with seed 17. The cross-host carryover subset used in sensitivity checks contains 20 rows (10 seeds; models Mistral-7B and Qwen2.5-14B). Host-delta group diagnostics are summarized in Table 7; CI-collapse diagnostics remain in the archived analysis bundle. All artifacts are frozen in a versioned reviewer bundle, including core/probe records, analysis outputs, and table regeneration scripts.

We also preserve a replay contract for each analysis artifact: deterministic run identifiers, host labels, seed provenance, and regeneration scripts that rebuild tables and figures from records. This supports open review by making release-policy outputs traceable to raw run rows and deterministic transforms.

### 4.5 Operational Assumptions

The protocol assumes a shared release policy with explicit human accountability at the promote/block boundary. Automated gates can decide "eligible for promotion," but final promotion authority still sits with the release owner. This is important because stop semantics, parity thresholds, and pass thresholds encode risk tolerance, not universal truth. In practice, teams should version these policies alongside code and model edits so historical release decisions remain auditable under the policy active at the time of release.

## 5 RESULTS

### 5.1 Finding 1: Seed Repetition Saturates in This Matrix

Across all configurations in this matrix, adding seed repetitions beyond the first did not change release decisions (mean seed-stop $n^* = 1.0$), as shown in Figure 2 and Table 2. This also holds within the limited controlled stochasticity probe, indicating that additional seed depth adds little decision value in this dataset. Artifact readiness checks reported zero error-level mismatches in core and augmented bundles, so this result is not explained by data-integrity confounds.

This finding has direct cost implications: compared with fixed 10-pass seed evaluation, seed-stop reduces seed-pass units by 90% and measured GPU-hours by about 90%. In practical terms, seed

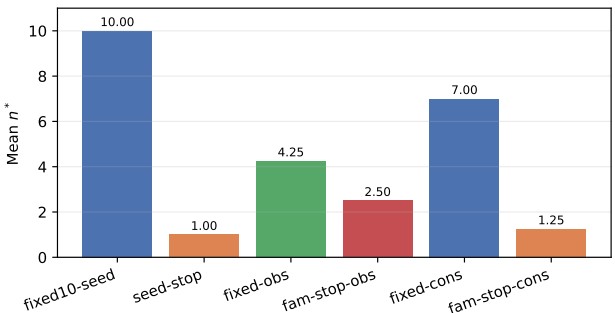

**Figure 2: Stop-depth comparison across fixed baselines and stop-rule variants.**

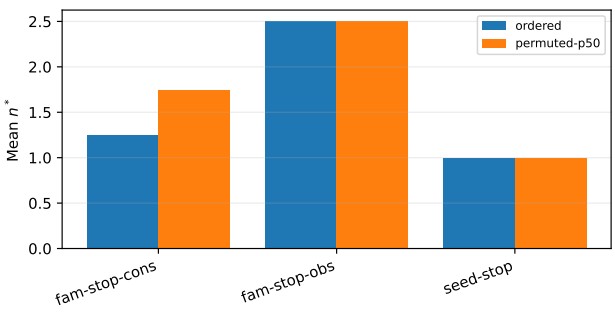

**Figure 3: Ordered vs permuted median stop depth (order-sensitivity check).**

depth beyond the first pass behaved as an accounting cost, not an information gain, for promote/block decisions in this matrix.

## 5.2 Finding 2: Edit-Family Diversity Remains Informative

Unlike seed depth, edit-family depth continues to change decisions. In the complete core matrix, family-stop remains at $n^* = 3.0$ under both semantics. In the augmented matrix, semantics diverge: observed coverage gives $n^* = 2.5$, while conservative missing-as-fail gives $n^* = 1.25$ (Figure 3; Table 3; Appendix Table 9). Stratification explains the shift: core-only groups remain at $n^* = 3.0$, while probe-only groups are single-family slices.

Order-sensitivity checks reinforce this interpretation. Under observed semantics, ordered and permuted medians agree. Under conservative semantics, ordered means are lower than permuted medians, showing that missingness policy and ordering interact. The cost tradeoff mirrors this behavior. Observed family-stop reduces pass units and GPU-hours by roughly 41–42% relative to fixed full observed coverage, while conservative semantics produce larger reductions that partly reflect policy strictness rather than empirical convergence.

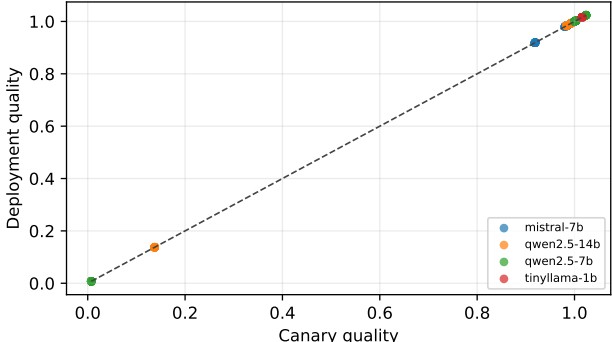

**Figure 4: Canary-to-deployment quality alignment (340 runs).**

## 5.3 Finding 3: Host-Parity Gating Changes Promotion Outcomes

Within the H100/H200 pairing studied here, host parity materially changes promotion decisions near strict boundaries. At $\epsilon = 0.001$ in the augmented matrix, admissibility is 88.2%, and 20 candidates that would pass under no-parity baselines are blocked by parity gating (Figure 5; Table 4). In the core matrix at the same threshold, admissibility is 93.3% with 10 blocked promotes.

Threshold scans show clear boundary effects. At pass threshold 0.95, host decision flips are zero; at stricter thresholds, flips appear (Figure 6; Table 5). Flip hotspots are concentrated in stochastic quantized runs and selected edit families rather than uniformly spread across the matrix (Table 6). The threshold value 1.001 in the scan is intentional as a boundary stress point for quality-ratio policies where scores can exceed 1.0.

Host-delta diagnostics are mostly small but non-zero. Table 7 consolidates all 17 edit/model groups into one view and uses manifest identifiers for the edit families. In those labels, the two quant_rtn_* entries denote RTN quantization settings, the LoRA-like rank-8 entry denotes the adapter-style edit, the runtime promptwrap entry denotes prompt wrapping, the inference-config entry denotes decoder/runtime perturbation, and the stochastic-sampling plus external-TinyLlama entries are the two probe-only families. Four groups show non-zero host deltas and 13 remain zero across all seeds; all four non-zero groups are Mistral-7B configurations. A plausible explanation is interaction between stricter boundary settings and stochastic/quantized Mistral runs, where small host-level numeric differences are more likely to cross the decision boundary; this is a workload-level hypothesis rather than an architectural claim.

Targeted probes sharpen the interpretation rather than overturn it. The external-family probe expands model-family coverage while remaining parity-stable. The canary/deployment scatter in Figure 4 lies on the identity line for this dataset; from the same 340 calibration records used for that plot, mean, median, P95, and max of $|c-d|$ are all 0.0. The stochastic probe increases host-offset pressure and boundary flips but does not make seed-depth stopping informative.

The detailed quantitative tables for Findings 1–3 are listed below for auditability and reproducibility.

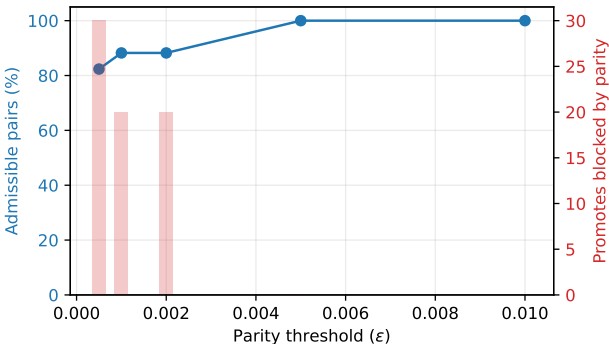

**Figure 5: Parity threshold transition and blocked-promote impact.**

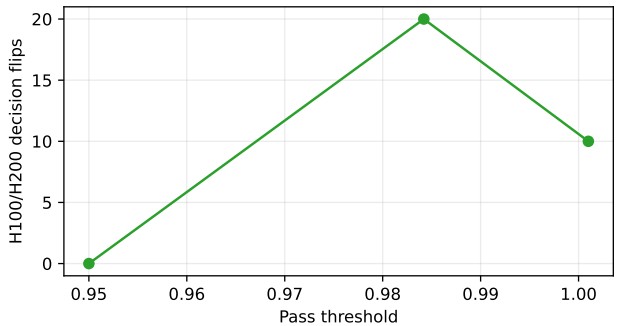

**Figure 6: Host decision flips across pass thresholds.**

**Table 2: Policy baseline comparison with pass-units and measured GPU-hours.**

| Policy | Mean $n^*$ | Reduction (%) | GPU-hours | GPU-hour red. (%) |
|---|---|---|---|---|
| Fixed 10-pass (seed) | 10.00 | 0.00 | 17.77 | 0.00 |
| Seed stop rule | 1.00 | 90.00 | 1.78 | 89.97 |
| Fixed full edit-set (observed) | 4.25 | 0.00 | 17.77 | 0.00 |
| Family stop rule (observed) | 2.50 | 41.18 | 10.26 | 42.25 |
| Fixed full edit-set (conservative) | 7.00 | 0.00 | 17.77 | 0.00 |
| Family stop rule (conservative) | 1.25 | 82.14 | 0.30 | 98.32 |

**Table 3: Family-stop stratification (all/core/probe slices).**

| Slice | Groups | Mean sequence length | Mean $n^*$ |
|---|---|---|---|
| all | 8 | 4.25 | 2.50 |
| core only | 6 | 5.00 | 3.00 |
| probe only | 4 | 1.00 | 1.00 |

## 5.4 Sensitivity Interpretation

Two sensitivity results are central to interpretation. First, missingness semantics change family-stop estimates in augmented evidence; this is a policy effect, not a numerical artifact, and should be reported explicitly in release decisions (Appendix Table 9). Second,

**Table 4: Host-parity transition with symmetric no-parity baselines.**

| Slice / $\epsilon$ | Adm. (%) | Blk H100 | Blk H200 | Blk Any |
|---|---|---|---|---|
| Augmented 0.0005 | 82.4 | 30 | 30 | 30 |
| Augmented 0.0010 | 88.2 | 20 | 20 | 20 |
| Augmented 0.0020 | 88.2 | 20 | 20 | 20 |
| Augmented 0.0050 | 100.0 | 0 | 0 | 0 |
| Augmented 0.0100 | 100.0 | 0 | 0 | 0 |
| Core 0.0005 | 86.7 | 20 | 20 | 20 |
| Core 0.0010 | 93.3 | 10 | 10 | 10 |
| Core 0.0020 | 93.3 | 10 | 10 | 10 |
| Core 0.0050 | 100.0 | 0 | 0 | 0 |
| Core 0.0100 | 100.0 | 0 | 0 | 0 |

**Table 5: Decision-flip scan across pass thresholds (H100 vs H200).**

| Pass threshold | Host decision flips |
|---|---|
| 0.9500 | 0 |
| 0.9842 | 20 |
| 1.0010 | 10 |

**Table 6: Flip hotspots by threshold (edit-level decomposition).**

| Threshold | Top edit family | Flips | Flip rate (%) |
|---|---|---|---|
| 0.9500 | stochastic_sampling_quant8 | 0 | 0.0 |
| 0.9842 | stochastic_sampling_quant8 | 10 | 100.0 |
| 1.0010 | lora_like_rank8_eps1e-3 | 10 | 33.3 |

threshold coupling shows that parity and pass thresholds govern different failure modes: parity controls transfer admissibility, while pass thresholds control within-host strictness (Tables 4 and 5). Treating them as a single dial hides decision tradeoffs.

This decision-convergence framing also differs from sequential probability ratio test (SPRT) style stopping [23]. SPRT assumes explicit stochastic models and error-rate targets for hypothesis decisions, while our setting uses deterministic policy semantics over finite release matrices. The stopping objective is therefore governance convergence of promote/block outcomes, not inferential acceptance/rejection against a probabilistic null.

## 5.5 Decision-Stage Walkthrough

The protocol can be read as a staged release decision process:

(1) **Artifact readiness.** Block analysis if required rows contain error-level identity, parsing, or metric issues.

(2) **Evidence sufficiency.** Evaluate seed-stop and family-stop under declared semantics to decide whether additional passes are justified.

(3) **Promotion control.** Apply pass threshold and host-parity admissibility, then compare blocked-promote counts against symmetric no-parity baselines.

This separation avoids a common failure mode: collapsing data quality, convergence, and promotion risk into one threshold.

**Table 7: Host-delta groups by edit family and model.**

| Edit | Model | Mean $\Delta_{host}$ | P50 | P90 | Max | Status |
|------|-------|---------|-----|-----|-----|--------|
| stochastic_sampling_quant8 | mistral_7b | 0.0049 | 0.0049 | 0.0049 | 0.0049 | non-zero |
| lora_like_rank8_eps1e-3 | mistral_7b | 0.0024 | 0.0024 | 0.0024 | 0.0024 | non-zero |
| quant_rtn_8bit | mistral_7b | 0.0007 | 0.0007 | 0.0007 | 0.0007 | non-zero |
| quant_rtn_4bit | mistral_7b | 0.0003 | 0.0003 | 0.0003 | 0.0003 | non-zero |
| external_tinyllama_quant8 | tinyllama_1b | 0.0000 | 0.0000 | 0.0000 | 0.0000 | zero across seeds |
| inference_cfg_perturb | mistral_7b | 0.0000 | 0.0000 | 0.0000 | 0.0000 | zero across seeds |
| inference_cfg_perturb | qwen2.5_14b | 0.0000 | 0.0000 | 0.0000 | 0.0000 | zero across seeds |
| inference_cfg_perturb | qwen2.5_7b | 0.0000 | 0.0000 | 0.0000 | 0.0000 | zero across seeds |
| lora_like_rank8_eps1e-3 | qwen2.5_14b | 0.0000 | 0.0000 | 0.0000 | 0.0000 | zero across seeds |
| lora_like_rank8_eps1e-3 | qwen2.5_7b | 0.0000 | 0.0000 | 0.0000 | 0.0000 | zero across seeds |
| quant_rtn_4bit | qwen2.5_14b | 0.0000 | 0.0000 | 0.0000 | 0.0000 | zero across seeds |
| quant_rtn_4bit | qwen2.5_7b | 0.0000 | 0.0000 | 0.0000 | 0.0000 | zero across seeds |
| quant_rtn_8bit | qwen2.5_14b | 0.0000 | 0.0000 | 0.0000 | 0.0000 | zero across seeds |
| quant_rtn_8bit | qwen2.5_7b | 0.0000 | 0.0000 | 0.0000 | 0.0000 | zero across seeds |
| runtime_promptwrap_custom | mistral_7b | 0.0000 | 0.0000 | 0.0000 | 0.0000 | zero across seeds |
| runtime_promptwrap_custom | qwen2.5_14b | 0.0000 | 0.0000 | 0.0000 | 0.0000 | zero across seeds |
| runtime_promptwrap_custom | qwen2.5_7b | 0.0000 | 0.0000 | 0.0000 | 0.0000 | zero across seeds |

## 5.6 Comparison with Fixed Checklist Evaluation

Many production pipelines still apply a fixed checklist policy: run all configured seeds and all configured edits, then apply one pass threshold. Against that baseline, this study shows two decision-theoretic inefficiencies (Table 2). First, fixed seed depth can consume most evaluation budget while leaving decision outcomes unchanged. Second, fixed checklists can miss transfer-risk behavior if cross-host admissibility is not modeled as a separate gate. In this matrix, parity gating blocks candidates that would otherwise be promoted by host-local checks alone (Table 4). This means that checklist completeness and release robustness are not equivalent; explicit gate semantics are needed to connect evaluation effort to release risk.

## 5.7 Cross-Finding Synthesis

The synthesis question is practical: if a release team can fund only one additional unit of evaluation, which control lever has the highest expected probability of changing a promote/block decision in a defensible way? Answering this requires combining cost, decision stability, and transfer-risk behavior rather than reading each finding in isolation.

Across Findings 1–3, the evidence separates high-cost/low-yield and lower-cost/high-yield actions. Seed-depth expansion beyond the first pass has near-zero observed decision gain while consuming substantial runtime. Edit-family expansion remains decision-informative through the observed stop depth in the core matrix. Host-parity checks occur less frequently but can block boundary promotions that would pass host-local checks.

A useful normalization is *marginal decision yield*: the number of final promote/block changes attributable to an added control lever, divided by the added pass budget. Under this lens, seed-depth expansion beyond pass 1 has effectively zero yield in this matrix, because additional seeds do not alter terminal outcomes. By contrast, edit-family expansion has positive yield through observed stop

depth, and parity checks have low event frequency but high consequence because each blocked promote corresponds to a prevented cross-host inconsistency near decision boundaries.

This distinction suggests a workload-conditional two-budget policy for release governance and a provisional budget order for this matrix: satisfy integrity prerequisites first, spend discovery budget on breadth until family-stop stabilizes, then spend assurance budget on parity checks for boundary candidates near strict thresholds. Deeper seed repetition should be deprioritized unless workload-specific stochasticity shows persistent decision movement under additional seeds. In this dataset, that combination produces lower expected waste than fixed checklists while preserving controls that materially change release outcomes.

This rule is not universal. It can fail under workloads with materially higher stochastic spread, unseen host classes with larger numerical divergence, or policy settings that intentionally optimize for recall over release conservatism. For those contexts, teams should rerun the same synthesis with local thresholds and semantics instead of transferring constants from this dataset.

The synthesis also suggests a trigger-based escalation ladder for marginal budget allocation: keep breadth-first family expansion as default, trigger targeted parity checks only for candidates within a narrow score band around the decision threshold, and increase seed depth beyond one pass only when stochastic probes show persistent post-parity decision movement. This keeps additional compute tied to observed instability rather than fixed checklist depth. Table 8 is a synthesis table rather than a separate empirical result: it summarizes the observed marginal-cost / marginal-decision-impact tradeoffs behind the proposed policy ordering.

## 6 DISCUSSION

### 6.1 Operational Guidance

For deployment, teams should operationalize the staged workflow defined in Section 5.5 and the budget priorities derived in Section 5.7 as explicit CI policy rather than ad hoc run expansion. In practice

**Table 8: Synthesis of evaluation controls: marginal cost vs decision impact.**

| Control lever | Incremental cost profile | Observed decision impact | Primary risk addressed | Recommended policy action |
|---|---|---|---|---|
| Seed-depth expansion beyond pass 1 | High runtime per added pass | Near-zero in this matrix | Under-sampling stochasticity | Deprioritize; use only if instability appears |
| Edit-family expansion to observed stop depth | Moderate, front-loaded cost | Non-trivial until family-stop stabilizes | Blind spots across edit classes | Prioritize first for marginal budget |
| Parity threshold tightening near boundary | Low extra compute, higher policy strictness | Lower admissibility; blocks boundary promotes | Host-transfer mismatch risk | Tune after pass-threshold calibration |
| Fixed checklist baseline | High fixed cost regardless of signal | Mixes informative and redundant runs | Budget dilution and hidden transfer risk | Keep as reference, not primary policy |
| Staged gate policy (proposed) | Adaptive cost with early stopping | Preserves high-impact checks with lower waste | Integrity, convergence, and transfer-risk coupling | Use as default with declared semantics |

this means preserving separate artifacts for integrity outcomes, stop-depth outcomes, and parity outcomes so that data quality, evidence sufficiency, and transfer risk remain auditable as distinct decisions.

## 6.2 Cost and Threshold Interaction

The stop rules reduce cost materially in this matrix, but reductions are not semantically equivalent. Conservative missing-as-fail policies can appear cheaper because they trigger earlier blocking under incomplete coverage. Practitioners should therefore report stopping semantics explicitly. Threshold tuning also has two coupled controls: pass thresholds govern per-host strictness, while parity thresholds govern cross-host transfer tolerance. A practical sequence is to tune pass threshold for quality objectives first, then tune parity threshold for transfer-risk control.

Three recurring decision cases summarize the empirical pattern. Case A: parity blocks otherwise passing host-local candidates near strict boundaries; this is expected transfer-risk control, not contradiction. Case B: seed expansion leaves decisions unchanged in this matrix; budget can shift to coverage breadth. Case C: conservative missing-as-fail yields rapid convergence; teams should declare it as worst-case governance rather than empirical sufficiency.

These cases motivate reporting discipline: each release policy result should disclose semantic choice (observed vs conservative), ordering policy for family-stop trajectories, and baseline definition for parity comparisons.

## 6.3 LLMOps Implications

Publicly described LLMOps workflows often emphasize experiment throughput and aggregate score dashboards [9], but they under-specify decision semantics for release gating. The evidence here suggests that release governance needs explicit contracts for artifact readiness, stopping depth, and cross-host admissibility. Without those contracts, similar scoreboards can still produce different promotion outcomes across teams.

The applied implication is not to replace existing LLMOps tooling, but to add a policy layer on top of it. Because FM vendor release processes are rarely documented with comparable stopping/parity detail, the novelty claim here is not that industry lacks such controls,

but that the paper makes those semantics explicit and auditable. Teams can keep existing evaluation runners and metric collectors while adopting explicit gate semantics and sensitivity disclosures as release requirements. This aligns with broader accountability patterns in model documentation and risk-management frameworks [14, 21]. The policy layer is portable in principle to stacks that emit deterministic run identifiers, host-tagged outputs, and typed integrity diagnostics, but this paper does not empirically validate multiple runners.

This portability also matters for teams that are not specialists in hardware-level reproducibility. With reusable CI gate templates, they can obtain bounded release decisions without requiring every contributor to reason directly about GPU numeric drift and threshold interactions.

## 6.4 Adoption Guidance

An implementation pattern that follows from this work is to separate CI orchestration into three jobs: (1) integrity validation and manifest locking, (2) decision-stability evaluation with early stopping checks, and (3) parity-gated promotion simulation. Each job should emit typed artifacts that downstream jobs consume, rather than re-deriving state from logs. This reduces hidden coupling between evaluation scripts and release dashboards.

A second pattern is role separation. Model engineers own edit construction and baseline selection; release engineers own gate thresholds and admissibility policy; approvers review policy outputs with sensitivity disclosures attached. This reduces the chance that threshold adjustments are made ad hoc under release pressure without traceability.

For reproducible review, each release-policy table should include six declarations: artifact integrity status, stop-rule semantic choice, ordering policy for family-stop, pass threshold, parity threshold, and no-parity baseline definition. The protocol is easiest to adopt when teams also define escalation rules before running evaluations. In our setting, three conditions triggered mandatory human escalation: integrity errors on required rows, parity-block events above a policy limit, and divergence between observed and conservative stop semantics beyond a configured tolerance.

A lightweight adoption checklist follows directly from the evidence:

(1) lock release policy metadata (thresholds, semantics, ordering) in version control;
(2) run integrity checks before any aggregate metric computation;
(3) execute a short seed-depth pilot to detect early saturation;
(4) spend remaining budget on edit-family breadth and targeted probes;
(5) enforce parity admissibility before final promotion;
(6) publish sensitivity disclosures with every release decision table.

Items 2 and 5 are direct gate requirements reinforced by the reported evidence; items 1, 3, 4, and 6 are deployment guidance derived from the same release-governance logic. The checklist is intentionally procedural and targets consistent decision quality under time-constrained release operations.

## 6.5 Operational Failure Modes and Mitigations

The evidence also clarifies where release pipelines fail when gate semantics are implicit. The first failure mode is *silent integrity drift*: summary metrics are computed from mixed or malformed rows, and downstream discussions focus on thresholds instead of artifact validity. The mitigation is to make integrity status a hard prerequisite and include issue-class counts in every release report.

The second failure mode is *false confidence from early convergence*. Under conservative missing-as-fail semantics, convergence can appear immediate even when observed coverage remains decision-informative. This mirrors a known risk in flaky-test settings, where apparently stable outcomes can hide policy-sensitive instability [10]. The mitigation is dual reporting: publish observed and conservative results side-by-side, with explicit ordering policy and sequence length.

The third failure mode is *host-local promotion bias*. Teams often approve releases from single-host evaluations, then discover deployment instability on a different host class. In this study, parity gating prevented that outcome for boundary cases. The mitigation is to treat host parity as a first-class promote gate and to report blocked-promote counts against symmetric baselines rather than one host-only baseline.

The fourth failure mode is *policy drift under schedule pressure*. Thresholds and stop semantics are adjusted ad hoc during a release crunch, making later audit impossible. The mitigation is governance hygiene: version policy metadata, require change justification, and include policy hashes in release artifacts so policy changes are as traceable as model edits.

## 7 THREATS TO VALIDITY

**Construct validity.** Endpoints are policy outputs (promote/block, admissibility, stop depth), not user-facing task success. The study therefore supports release-governance claims, not downstream task-value guarantees.

**External validity.** Core evidence is broad enough for multi-edit/multi-model analysis but still finite. Probe arms are targeted stress tests, not full Cartesian expansions. Results are expected to transfer at the policy level, while numeric thresholds remain workload-dependent.

**Order dependence.** Stop-depth metrics depend on pass ordering. We mitigate via permutation summaries and report order-sensitive variants explicitly. Teams using different family-ordering policies may observe different stop depths and should report ordering policy with results.

**Toolchain dependence.** Single-toolchain measurements may bias results. Mitigation uses deterministic manifests and replayable artifact contracts [16].

## 8 LIMITATIONS AND FUTURE WORK

This study prioritizes release-governance decisions and does not claim downstream task correctness. The matrix spans multiple models, edits, and two hardware classes, but it is still finite and centered on one toolchain. Two practical extensions are high priority: broader stochastic presets for non-trivial seed-stop behavior beyond controlled probes, and a wider cross-device matrix (for example

mixed host classes per stage) to test whether the parity-threshold transition generalizes beyond this dataset.

Numeric thresholds reported here are workload-specific, not universal constants. In particular, parity cutoffs and boundary-band widths depend on metric scale, decoding setup, and hardware/runtime stack. We therefore expect policy structure (integrity → stop depth → parity gating) to transfer more reliably than exact threshold values. A direct replication path for new teams is to preserve the same semantic split (observed vs conservative), rerun permutation sensitivity checks, and re-fit parity cutoffs against local blocked-promote tolerances before operational rollout.

## 9 CONCLUSION

When missingness semantics and host baselines are handled explicitly, the evidence in this single-toolchain H100/H200 matrix is consistent: seed repetition saturated quickly here, edit-family diversity remained decision-informative, and host parity could change promotion outcomes near strict boundaries. The practical release rule is therefore explicit and staged: integrity first, diversity-aware stopping second, parity-gated promotion third. These findings suggest that FM release engineering benefits from explicit stopping and parity policies, a governance layer that current LLMOps abstractions still under-specify.

# A APPENDIX

## A.1 Order-Sensitivity Summary

**Table 9: Order-sensitivity summary for $n^*$ under pass permutations.**

| Axis/policy | Groups | Ordered mean | Permuted P50 mean |
|---|---|---|---|
| Seed stop | 34 | 1.00 | 1.00 |
| Family stop (obs.) | 8 | 2.50 | 2.50 |
| Family stop (cons.) | 8 | 1.25 | 1.75 |

## ACKNOWLEDGMENTS

Generative AI assistants were used to improve draft wording and edit prose/table phrasing. All experiments, code changes, calculations, and final technical claims were verified by the human authors.

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
